# A Study on the Synbiotic Composition of *Bifidobacterium bifidum* and Fructans from *Arctium lappa* Roots and *Helianthus tuberosus* Tubers against *Staphylococcus aureus*

**DOI:** 10.3390/microorganisms9050930

**Published:** 2021-04-26

**Authors:** Svetlana A. Evdokimova, Vera S. Nokhaeva, Boris A. Karetkin, Elena V. Guseva, Natalia V. Khabibulina, Maria A. Kornienko, Veronika D. Grosheva, Natalia V. Menshutina, Irina V. Shakir, Victor I. Panfilov

**Affiliations:** 1Department of Biotechnology, Faculty of Biotechnology and Industrial Ecology, D. Mendeleev University of Chemical Technology, Miusskaya Sq., 9, 125047 Moscow, Russia; s.a.evdokimova@gmail.com (S.A.E.); veravenice@yandex.ru (V.S.N.); ernestine2007@yandex.ru (N.V.K.); vgrosheva@muctr.ru (V.D.G.); ivshakir@muctr.ru (I.V.S.); vip@muctr.ru (V.I.P.); 2Department of Cybernetics of Chemical Technological Processes, Faculty of Digital Technologies and Chemical Engineering, D. Mendeleev University of Chemical Technology, Miusskaya Sq., 9, 125047 Moscow, Russia; eguseva@muctr.ru (E.V.G.); chemcom@muctr.ru (N.V.M.); 3Federal Research and Clinical Center of Physical-Chemical Medicine of Federal Medical Biological Agency, 119435 Moscow, Russia; kornienkomariya@gmail.com

**Keywords:** *Bifidobacterium bifidum*, fructans, burdock roots, Jerusalem artichoke tubers, synbiotics, *Staphylococcus aureus*, growth inhibition model, coculture

## Abstract

A number of mechanisms have been proposed explaining probiotics and prebiotics benefit human health, in particular, probiotics have a suppression effect on pathogen growth that can be enhanced with the introduction of prebiotics. In vitro models enhanced with computational biology can be useful for selecting a composition with prebiotics from new plant sources with the greatest synergism. Water extracts from burdock root and Jerusalem artichoke tubers were purified by ultrafiltration and activated charcoal and concentrated on a rotary evaporator. Fructans were precipitated with various concentrations of ethanol. *Bifidobacterium bifidum* 8 VKPM AC−2136 and *Staphylococcus aureus* ATCC 43300 strains were applied to estimate the synbiotic effect. The growth of bifidobacteria and staphylococci in monocultures and cocultures in broths with glucose, commercial prebiotics, as well as isolated fructans were studied. The minimum inhibitory concentrations (MICs) of lactic and acetic acids for the *Staphylococcus* strain were determined. A quantitative model joining the formation of organic acids by probiotics as antagonism factors and the MICs of pathogens (as the measure of their inhibition) was tested in cocultures and showed a high predictive value (R^2^ ≥ 0.86). The synbiotic factor obtained from the model was calculated based on the experimental data and obtained constants. Fructans precipitated with 20% ethanol and *Bifidobacterium bifidum* have the greater synergism against *Staphylococcus*.

## 1. Introduction

The suppression effect of probiotics against pathogens is the basis for health maintenance of not only the gastrointestinal tract but the whole organism. Antagonism allows these beneficial microorganisms to be classified as one of the most important components of functional foods. The concept of probiotics and prebiotics is becoming more widespread every year, which, in particular, is associated with the need to fight against antibiotic-resistant pathogens.

Most often, microorganisms of the species *Escherichia coli*, *Klebsiella pneumoniae*, *Staphylococcus aureus*, *Streptococcus pneumoniae*, and *Salmonella *spp. are considered antibiotic-resistant [1]. The strains of *Staphylococcus aureus* are capable of biofilm formation, which significantly increases their antibiotic resistance and colonization activity of the large intestine. The biofilm is the most resistant form when *Staphylococcus* is able to survive, even in the presence of vancomycin [2]. *Staphylococcus aureus* causes serious complications in the treatment of many diseases (causes secondary hospital infections) that significantly increase the duration of treatment. For example, the treatment of metastatic disease in the presence of *Staphylococcus* is extended to six weeks, whereas adults with uncomplicated *S. aureus* bacteremia only require two weeks of antimicrobial treatment [3]. The diseases caused by *S. aureus* include neonatal purulent parotitis, various acute infections of the skin and skin structure, severe respiratory diseases such as bronchiectasis and chronic obstructive pulmonary disease [4] and urinary tract infections (caused in 5–10% of cases *Staphylococcus aureus*) [5]. *Staphylococcus aureus* has been reported as the main cause of enterocolitis and diarrhea in antibiotic-treated patients [6]. The interest in the search for new antibiotics with a new mode of action to suppress this pathogen, such as phenyl thiazoles, oxazolidinones, benzimidazoles, and chalcones, is ongoing [7]. However, this approach does not exclude the possibility that bacteria can acquire new mechanisms to counteract antimicrobial substances and are not the only possible solution to the problem.

The ability of probiotics to inhibit the growth of pathogens and food contaminants corresponds to the production of short-chain fatty acids (SCFAs), bacteriocins, and other antimicrobial compounds. They are also able to compete with unwanted microbes for adhesion sites on the intestinal epithelium and for nutrients [8]. Prebiotics stimulate probiotic growth and the formation of SCFAs, thus enhancing the beneficial effects of probiotics. Probiotics and prebiotics could be considered as possible treatment options for cognitive impairment [9]. Thus, the combined use of probiotics and prebiotics in synbiotic compositions could be the most effective approach to modulate the intestinal microbiota and to prevent and provide complex treatment for various diseases including bacterial superinfections. Studies involving the feeding of synbiotic yogurt based on lactobacilli and soybean fructooligosaccharides (FOS) to mice in a diet enriched with cholesterol have demonstrated decreases in the levels of total serum cholesterol and triglycerides with synbiotic yogurt showing better antioxidant activity [10].

As part of the search for new substances with prebiotic properties, a great amount of attention has been directed to oligo- and polysaccharides isolated from natural sources such as Jerusalem artichoke, chicory, rye, milk, honey, onion, barley, and salsify [11]. For example, it has been shown that *Lactobacillus paracasei*, in combination with rice bran extract, reduces the growth rate of *Salmonella typhimurium* more effectively than without it [12]. The effect of probiotic stimulation by different types of honey has been demonstrated by researchers on a number of strains of bifidobacteria and lactobacilli in vitro. The ability to fermenting honey oligosaccharides was revealed is strain-specific, that reaffirming one of the most important properties of prebiotics affecting their effectiveness. [13]. Increased antimicrobial activity was observed in *L. lactis* subsp. *lactis*, *L. reuteri*, and *P. acidilactici* grown in media with β-galactooligosaccharides (GOS) from barley as the sole carbon source compared with glucose. In particular, β-GOS enabled sustained *L. lactis* subsp. *lactis* growth in the exponential phase, resulting in an increase of approximately 25% in nisin Z production [14]. Moreover, plant polyphenols have recently been recognized as noncarbohydrate prebiotics. The coencapsulation of *Bifidobacteria* with green tea phenols enhanced their stability in a simulated gastrointestinal environment and under refrigerated conditions [15].

When plant extracts are included in synbiotic compositions, it is not only possible to obtain a synergic effect between probiotics and prebiotics, but there is also a wide range of beneficial properties from other compounds contained in the plant. Many medicinal plants contain their own antimicrobial substances that inhibit the growth of pathogens, for example, plant extracts and phytochemicals, as antimicrobial and antibiofilm agents, which appears to be a promising, cost-effective, and ecofriendly approach against *Salmonella* species [16]. In a DSS-induced colitis mice model, water-soluble polysaccharides from *Arctium lappa* significantly increased the concentrations of Firmicutes, *Ruminococcaceae*, *Lachnospiraceae* and *Lactobacillus* while notably inhibiting the levels of Proteobacteria, *Alcaligenaceae*, *Staphylococcus*, and Bacteroidetes and ameliorating the dysregulation of pro-inflammatory and anti-inflammatory cytokines, confirming its effect in protecting mice from colitis [17]. Arktigenin and tannin contained in the roots of *Arctium lappa* have anti-inflammatory and antitumor effects. In addition, tannin exhibits immunomodulatory properties [18]. Phenolic acids from plants contribute to an overall health improvement primarily by virtue of their antioxidant, anti-inflammatory, antimicrobial, antimutagenic, hypoglycemic, and antiplatelet aggregating activities [19]. The synergism of purified fructans from burdock roots or Jerusalem artichoke tubers and probiotics, especially bifidobacteria, is in interest to allow wider understanding of the mechanisms of health benefits. However, few related studies have been conducted.

Thus, through the rational combination of probiotic bacteria and plant extracts containing prebiotics, the greatest suppression of a particular pathogen can be achieved. In this case, the synergic effect will be aimed at increasing the production of inhibitory substances by probiotics, and the prebiotic substrate should be unconsumable or at least poorly consumable by the pathogen. As a comparative criterion to assess the effectiveness of such compositions, the previously proposed synbiotic factor (SF) can be used, which shows how many times the specific growth rate of the pathogen will decrease as a result of the inhibitory effect of the prebiotics on metabolic products [20]. The aim of our study is to determine the most effective composition of *B. bifidum* and various fractions of fructans from burdock and Jerusalem artichoke against *Staphylococcus* basing on the validated quantitative competition model of probiotics and pathogen.

## 2. Materials and Methods

### 2.1. Plant Raw Materials

Tubers of the Jerusalem artichoke (*Helianthus tuberosus* L.), collected in the 2019 harvest and obtained from markets in Moscow city, were used as raw plant materials for the isolation of prebiotic substances. Burdock roots (*Arctium lappa* L.) were purchased as medicinal plant materials according to the National Pharmacopoeia of the Russian Federation (Monograph 2.5.0025.15).

### 2.2. Isolation of Fructans

In order to exclude a possible negative effect of noncarbohydrate substances on probiotics, the aqueous extraction of raw materials was carried out, followed by the separation of impurities. Jerusalem artichoke tubers and Burdock roots were washed and milled to 0.5–1 mm particles. Crushed raw materials were extracted with distilled water (solid dry matter to solvent ratio 1:12) at 75 °C during 30 min twice. The pulp was separated by vacuum filtration, and the extract was purified using ultrafiltration (polysulfone, molecular weight cut-off 20 kDa). To remove the colored impurities (polyphenolic components, furfural etc.), the extracts were clarified with activated charcoal, followed by separation of the coal by vacuum filtration [21]. The clarified extract was concentrated using the Hei-Vap Advantage rotary evaporator (model 561−01110−00 with glass set G1, Heidolph, Germany) giving a total carbohydrate content of 150–200 g·L^−1^. The concentrating was carried out at temperatures not exceeding 45 °C. To separate the carbohydrate fraction, the concentrate was poured into bottles, and ethanol was added to final concentrations of 20 and 80% v/v. In this way, the polysaccharides were precipitated with higher and lower degrees of polymerization, respectively [22]. The bottles were sealed thoroughly, mixed by shaking, and incubated for 2–3 days at 4–8 °C. The precipitates were separated by centrifugation for 15 min at 5000 rpm and freeze-dried. The total carbohydrate assay was carried out on the samples using the modified Fehling method after hydrolysis of the samples with 10% TCA for 40 min in a boiling water bath.

### 2.3. Bacterial Strains and Culture Conditions

The strain *Bifidobacterium bifidum* 8 VKPM Ac−2136 was purchased from LLC AVAN (Moscow, Russia) and studied as a probiotic (Pr). *Staphylococcus aureus* ATCC 43300, purchased from the American Type Culture Collections, was used as a pathogen test strain (Pt).

The inoculum preparation and fermentations were carried out in a medium prepared in accordance with Rossi et al. [23] with some modifications. The composition of the carbohydrate-free medium was as follows (in grams per liter): Casein tryptone (Difco Laboratories, Detroit, MI, USA), 10; yeast extract (Springer, Maisons-Alfort, France), 7,6; meat extract (Panreac, Barcelona, Spain), 5; ascorbic acid (AppliChem, Darmstadt, Germany), 1; sodium acetate, 1; (NH_4_)_2_SO_4_, 5; urea, 2; MgSO_4_·7H_2_O, 0.2; FeSO_4_·7H_2_O, 0.01; MnSO_4_·7H_2_O, 0.007; NaCl, 0.01; Tween−80, 1; and cysteine, 0.5 (with the pH adjusted to 7.0). Oligofructose (Orafti^®^ P95, BENEO-ORAFTI, Tienen, Belgium) (FOS) and lactulose (dry «Lactusan», LLC Felicite holding, Moscow, Russia) were applied as control prebiotics. The carbohydrates were solved in distilled water. The carbohydrate-free medium and carbohydrate solutions were sterilized separately at 115 °C for 30 min. The carbohydrate solutions were aseptically added to carbohydrate-free medium before inoculation to obtain a concentration of 10 g L^−1^.

The studies on bacterial growth kinetics (inhibition of *Staphylococcus* and co-culture competitions assays) were performed in a sealed vessel with two branches one of which (sampling branch) reached the bottom and the other one (inner branch) was not. Each of the branches had a membrane autoclavable vent filter (Midisart 2000 PTFE, 0.2 μm, Sartorius, Goettingen, Germany) and clamps. The vessels were filled with carbohydrate-free medium and sterilized as described above. The inoculates were obtained in the sealed vessel too. Moreover, the same carbohydrate was applied for inoculate preparation and batch fermentation. The overnight cultures (approx. 16 h) of the bifidobacteria and *S. aureus* strains were used to inoculate the vessels. The optical density (OD) of inoculates was measured and quantity of inoculate for each strain was calculated considered than 0.5 units of OD corresponded to 5 × 10^8^ CFU mL^−1^ of *Staphylococcus* and 3.5 × 10^8^ CFU mL^−1^ of bifidobacteria. Cells were washed and resuspended in a sterile phosphate-buffered saline (PBS) and the vessels were inoculated. The vessels were filled with N_2_ (extra pure) through the sampling branch immediately after inoculation. Fermentation was carried out at 37 °C with shaking (180 rpm) and pH was not maintained. The samples were taken aseptically hourly. The filter was removed from the sampling branch and N_2_ was pushed through the inner branch at about 0.5 L per min to press out the sample.

Fermentation with plant extracts was carried out in conical flasks with shaking (120 rpm) at 37 °C under anaerobic conditions (2% CO_2_, 98% N_2_) in a CB−210 CO_2_ incubator (Binder, Tuttlingen, Germany). The pH was not maintained. The inoculates were prepared in the same conditions in FOS-contained media overnight. The volumes of inoculates added were calculated as described above. Cells were preliminary washed and resuspended in sterile PBS. The samples were taken after inoculation and then the flasks were placed in CO_2_ incubator. The final samples moments of fermentation. Bacterial counts, pH and concentrations of acids were measured in all the samples.

### 2.4. Enumeration of Bacterial Growth

The bacterial counts of each measure were carried out in triplicate. Tenfold serial dilutions of the analyzed cell suspensions in sterile PBS were prepared. *S. aureus* colonies were counted by plating on selective medium mannitol salt agar (MSA) [24] at 37 °C in air. Bifidobacteria colonies were counted on BFM medium [25] composed of the following (g·L^−1^): Peptone, 10; sodium chloride, 5.0; lactulose, 5.0; cysteine hydrochloride, 0.5; riboflavin, 0.01; yeast extract, 7; meat extract, 5; starch, 2; thiamine chloride, 0.01; and lithium citrate, 3.3. The pH was adjusted to 5.5 by adding propionic acid (5 mL L^−1^). The plates were incubated under anaerobic conditions provided by the BD GasPak™ anaerobic container system at 37 °C. The specific growth rate was calculated as the slope of log_10_ of the bacterial count in the exponential phase to time.

### 2.5. Organic Acid Measurement

The concentrations of lactic and acetic acids were measured by HPLC, as described previously [26] with some modifications. The chromatographic evaluation was performed using an Agilent 1220 Infinity chromatographic system with refractometric detection (Agilent, Santa Clara, CA, USA) on an Agilent Hi-Plex H column (250 × 4.6 mm). The samples were centrifuged at 12,000 rpm for 15 min. The supernatant was filtered through an 0.45 μm cellulose acetate membrane (HAWP, MF-Millipore, St. Louis, MO, USA). Chromatography was performed at 50 °C. The mobile phase was 0.002 M H_2_SO_4_. The elution was carried out isocratically at a flow rate of 0.3 mL·min^−1^. The refractometric detector was set to a temperature of 50 °C, and the injection volume was 3 µL. For calibration, the standard solutions of acids and carbohydrates (10 mg·mL^−1^) were prepared with subsequent dilution in the mobile phase to obtain a concentration range from 1 to 10 mg·mL^−1^. The determination of conformity to the substance was carried out using a similar retention time on the column, and the concentration was defined by calculating the square of the chromatographic peak using the external standard method.

### 2.6. Assay of Carbohydrates (Oligosaccharides) by High-Performance Capillary Electrophoresis (HPCE)

The determination of oligosaccharides in the samples was carried out using the Capel−105M capillary electrophoresis system (Lumex^®^, Saint-Petersburg, Russia) equipped with quartz capillary (length 75 cm, internal diameter 50 μm) according to the method described by Andersen et al. [27] and Arentoft et al. [28] with some modifications. Before the analysis, the samples were centrifuged at 8500 rpm for 15 min to separate the bacterial biomass. Proteins were removed on 3 kDa centrifugal filter units Amicon^®^ Ultra−4 (Merck Millipore Ltd., Carrigtwohill, Ireland) and filtrates were analyzed. A solution of 25 mM pyridine−2,6-dicarboxylic acid (dipicolinic acid), 170 mM NaOH and 0.5 mM tetradecyltrimethylammonium bromide (C_17_H_38_NBr) (TDTMAB) was used as a background electrolyte. The indirect photometric detection of oligosaccharides was carried out at a wavelength of 254 nm.

### 2.7. The Quantitative Model of Probiotics and Pathogen Competition and Calculations

The competition model developed previously [20] was applied in the current study with some variations. The basic assumptions of the model are as follows: (a) The concentrations of substances produced by the pathogens do not affect probiotic growth; (b) the competition is related to metabolite formation by the probiotics, which reduces the specific growth rate of the pathogens; (c) the inhibitors formed suppress pathogen growth only; (d) the number of inhibitors is limited; and (e) there is no limitation on the substrate.

Thus, the system of equations for the coculture are as follows:(1)xPr=xPr0· expμPrmax·tIi=Pi=YPiX· xPr−xPr0=YPX·xPr0· expμPrmax·t−1μPt=μPtmax·fIdxPtdt=μPtxPt
where $x_{Pr}$ and $x_{Pt}\;$
are the concentration of probiotics (Pr) and the pathogen (Pt) count (CFU·mL^−1^) at the moment of fermentation t; $x_{{Pr}_{0}}$ is the probiotics count at the moment of inoculation (CFU·mL^−1^); $\mu_{{Pr}_{max}}$ and $\mu_{{Pt}_{max}}$
are the maximal specific growth rates of the probiotics and pathogens in h^−1^; $I_{i}$ ($P_{i}$) is the concentration of the inhibitor (metabolite); and YPiX is the yield of this metabolite/biomass.

The inhibitory effect of SCFAs on pathogen growth is described by the minimum inhibitor concentration (MIC) equation [29,30]:(2)$$\mu_{Pt}=\mu_{{Pt}_{max}}\cdot \;\frac{pH-pH_{min}}{pH_{opt}-pH_{min}}\cdot \left[1-{\left(\frac{\left[L\right]}{\;MIC_{L}}\right)}^{\alpha }\right]\left[1-{\left(\frac{\left[A\right]}{\;MIC_{A}}\right)}^{\beta }\right]$$
where [*L*] and [*A*] are the concentrations of nondissociated lactic and acetic acids, respectively, in mg·mL^−1^; MIC is the minimum inhibitory concentration in mg/mL; and *α* and *β* are constants.

The synbiotic factor (SF), which shows the number of times by which the specific growth rate of the pathogen will decrease at the end of the probiotic growth period, can be obtained from Equations (1) and (2):(3)SF=pH−pHmin pHopt−pHmin·1−YLxPr·xPrfinMICLα1−YAxPr·xPrfinMICAβ.

The fermentation of *S. aureus* monoculture has been developed at the mentioned conditions with the addition of different quantities of one of the acids (lactic acid or acetic acid). The specific growth rate was determined for each concentration of each acid by estimating the line of best fit. The constants of Equation (2) were calculated using regression analysis for the modification of Equation (2):(4)$$\mu_{Pt}=\mu_{{Pt}_{max}}\cdot \;\frac{pH-pH_{min}}{pH_{opt}-pH_{min}}\cdot \left[1-{\left(\frac{\left[L\right]}{\;MIC_{L}}\right)}^{\alpha }\right]$$
(5)$$\mu_{Pt}=\mu_{{Pt}_{max}}\cdot \;\frac{pH-pH_{min}}{pH_{opt}-pH_{min}}\cdot \left[1-{\left(\frac{\left[A\right]}{\;MIC_{A}}\right)}^{\beta }\right].$$

The growth curves of *B. bifidum* were calculated using exponential growth equation [31].

To simulate the growth of the *S. aureus* monoculture, the Verhulst equation was applied [32], as follows:(6)$$x_{{Pt}_{t}}=\frac{x_{{Pt}_{0}}\cdot K\cdot exp\left(r\cdot t\right)}{K+x_{{Pt}_{0}}\left[\exp\left(r\cdot t\right)-1\right]}$$
where *K* is the saturation level, meaning the maximal count of *S. aureus* in CFU·mL^−1^, and *r* is a constant corresponding to the maximum specific growth rate in h^−1^.

The specific growth rate of *S. aureus* was calculated in a few steps based on the equations written above: (1) The bifidobacteria count was obtained from the exponential growth equation; (2) the concentrations of inhibitors were calculated from the yields and bifidobacteria count; and (3) the *MIC* Equation (2) was applied to determine $\mu_{{Pt}_{t}}$. As the specific growth rate is a complex function of time, its diminution and the pathogen count were calculated by integrating the exponential growth equation in a narrow time interval (∆t = 1 h):
(7)$$x_{{Pt}_{t+1}}=x_{Pt}{}_{t}\cdot \;\exp(\mu_{{Pt}_{t}}\cdot ∆t).$$

### 2.8. Statistical Analysis

Each result represents the mean ± SD of three different experiments. The data were analyzed by the ANOVA test using MathLab software. Statistical significance was taken at *p* = 0.05.

## 3. Results

### 3.1. Determination of the Model Parameters for S. aureus Growth Inhibition by Lactic and Acetic Acids

The inhibition constants of *S. aureus* were determined separately for lactic and acetic acids. To bring the results closer to coculture fermentation conditions, the pH value was not maintained in this study. Therefore, the experimental data reflect the complex influence of the acid concentrations and pH. The comparison of the obtained patterns (Figure 1) and inhibition constants (Table 1) conducted with Equation (2) showed that acetic acid has the greatest inhibitory influence on *S. aureus* growth. For both of the regressions, R^2^ values were acceptable, so their prediction ability can be considered good.

### 3.2. Validation of the Model for the Coculture of S. aureus and B. bifidum Fermentation

Then, the set of experiments with the cocultures was carried out to compare the experimental data for the inhibition by pathogens with the data calculated using the MIC model. *S. aureus* and *B. bifidum* were cocultured in the medium containing prebiotics (FOS or lactulose) or glucose at a concentration of 10 g·L^−1^. Since the equations (7) include the value of the initial count of the pathogen, the initial count of *S. aureus* was additionally varied by 6 and 7 log(CFU·mL^−1^) to test the model. The joined experimental results are shown in Table 2. In all experiments the initial counts of bifidobacteria were approximately the same: 7.6–7.8 log(CFU·mL^−1^). The growth rate of bifidobacteria differed slightly (0.47–0.51 h^−1^). The increase in the initial count of staphylococci did not affect the yield of acids. However, bifidobacteria produced acids with different yields on various substrates. The highest of them were observed with FOS (0.60 pg CFU^−1^ for lactic acid and 0.56 pg CFU^−1^ for acetic acid, respectively). The production of lactic and acetic acids did not differ significantly from each other during growth on glucose and FOS; however, on lactulose, the yield of lactic acid was almost half the value of acetic acid.

According to the experimental data and the parameters previously obtained from the model, the growth curves of the *Staphylococcus* monoculture (Verhulst model) and coculture (MIC model) were calculated (Figure 2). When there was a visible lag-phase of pathogen growth, the calculation of changes in the count using the MIC model was conducted using the first point of population growth (Figure 2A–C,E). The initial (maximal) specific growth rate of the pathogen in coculture was calculated from the first points of the growth curve, as the concentrations of acids and their inhibitory effects during this period were minimal. The initial specific growth rates of *Staphylococcus* were higher on all carbohydrates at its lower initial count. A greater difference in growth rates was observed with growth on FOS (1.13 h^−1^ at lower inoculum dose of *Staphylococcus* and 0.81 h^−1^ with a higher initial count).

In all cases, the final count of *S. aureus* cells in coculture was less than in monoculture (Figure 2). At the same time, when the initial *Staphylococcus* population was 7 log(CFU mL^−1^), the final counts in monoculture and coculture differed slightly compared with a smaller initial count. It was interesting that the greatest difference in the final count of staphylococci in mono and cocultures was observed at a lower initial population on lactulose, and the smallest value was observed with a higher initial population on the same substrate.

Except for the case of a higher initial pathogen count on lactulose (Figure 2F), a decreased staphylococci count was observed at the final stage of growth. It should be noted that, in monoculture, *Staphylococcus* reaches approximately the same maximum count (8.6–8.8 log(CFU·mL^−1^)) regardless of the inoculum dose and substrate. At the same time, in coculture, *Staphylococcus* reached the maximum count on all substrates at a higher inoculum dose followed by a population decrease. On the other hand, at lower inoculum doses, lysis began before the maximum count was reached. Therefore, the effectiveness of the synbiotic composition is highly dependent on the initial inoculum dose of the pathogen.

Presumably, the decrease in the pathogen count is associated with the qualitative leaps of lactic and acetic acids actions from bacteriostatic to bacteriolytic conditions. These leaps occurs when acid concentrations exceed the definite MIC values and/or the pH value falls below the minimum pH (5.05 for the *S. aureus* strain studied). It should be noted that the MIC model implies the possibility of describing this area mathematically (if one of multiply is below zero). It should be noted that a variance between the experimental and calculated data was observed in the phase of count decrease. In addition, the calculated data did not reach the experimental peak value of the staphylococci count in any case (the maximum value of the staphylococci count, after which the count begins to decrease). It was interesting that with a higher inoculum dose of *Staphylococcus* (and lower inhibitory effect), the MIC model showed the best predictive ability (R^2^ = 0.99). In general, the MIC model describes the decrease in the specific growth rate of pathogen as a result of inhibitory acid production by probiotics accurately in all cases. The obtained R^2^ values for all variants were higher than 0.86.

### 3.3. Monocultures of B. bifidum and S. aureus Fermentation with Jerusalem Artichoke and Burdock Fructans

The burdock *(Arctium lappa* L.) roots and Jerusalem artichoke (*Helianthus tuberosus* L.*)* tubers were processed as described above to extract carbohydrates. Two fractions of burdock (Burd−20 and Burd−80) and Jerusalem artichoke (JA−20 and JA−80) fructans were obtained in the final stage by precipitation at ethanol concentrations of 20% v/v and 80% v/v, respectively. It was suggested that only fructans with a higher degree of polymerization (DP) would be precipitated at the lower ethanol concentration [22]. All samples were white (colorless in solution) and contained no less than 90% carbohydrates relative to dry matter.

To test the ability of the studied strains of probiotics and pathogens to consume the isolated fructans, monoculture fermentation was performed with the tested carbohydrates, control prebiotics (FOS), and nonprebiotics (glucose). Fermentation was carried out for 8 h, as the previously obtained growth curves showed that by this time, the *Staphylococcus* monoculture had reached the stationary phase. These results are presented in Table 3.

Of the two fractions of fructans, greater accumulation of bifidobacteria was observed during the fermentation on carbohydrates precipitated with 80% ethanol (9.47 log(CFU·mL^−1^) for Burd−80, and 9.21 for JA−80). These values are close to those obtained with the standard prebiotics, FOS (9.19 log(CFU·mL^−1^)) and higher than the final count of bacteria obtained on glucose (8.75 log(CFU·mL^−1^)). The formation of lactic and acetic acids on different substrates varied greatly. In these experiments, a great shift in the production of acids by bifidobacteria towards acetic acid was observed. Moreover, the accumulation of lactic acid was extremely low (less than 0.06 g L^−1^) on both fractions of burdock fructans (Burd−20 and Burd−80), Jerusalem artichoke fructans (JA−80), and glucose. The greatest level of accumulation of both acids was recorded during cultivation on FOS (0.56 g L^−1^ lactic and 1.53 g L^−1^ acetic acids). Among the two fractions from the same source, acid production was better on the carbohydrates with a higher DP. Based on the obtained data, capillary electrophoresis was carried out for the variants with the best acid production (FOS, Burd−20 or JA−20). The electrophoresis data (Figure 3) show that FOS contained large amounts of low molecular weight carbohydrates, while the fructans of burdock and Jerusalem artichoke were characterized by much smaller peak areas. It should be noted that this technique is not very informative for fructans with a DP of more than 7. In addition, in this case, capillary electrophoresis can be used as a method of semiquantitative analysis of the carbohydrate composition. A decrease in the peak areas of carbohydrates of various molecular weights was clearly visible, which indicates the consumption of these carbohydrates by bifidobacteria. For a more accurate assessment, further analysis of the isolated fructans with more sensitive methods (for example, NMR or High-performance anion-exchange chromatography with Pulsed amperometric detection) should be carried out.

No significant differences in the final count of *Staphylococcus* were observed on control and test substrates. In all cases, *Staphylococcus* reached the previously established final cell count (8.5–8.8 log(CFU·mL^−1^)). It should be noted that in the final samples of staphylococci monoculture, a large amount of lactic acid was found. Moreover, in all variants more lactic acid than bifidobacteria was produced on the same carbohydrate, and the highest value was observed on glucose (2.11 g L^−1^). The production of acetic acid on all substrates was small in comparison with that of lactic acid and it did not exceed the values obtained during the fermentation of probiotics.

The utilization of individual fractions of burdock and Jerusalem artichoke carbohydrates by the pathogens was confirmed by the electropherograms presented in Figure 3. Decreases in the areas of the corresponding peaks relative to the original nutrient media were noticeable on all substrates. However, it should be noted that *S. aureus* completely consumed the limited spectrum of substances from the mixture of homologues for all studied fructans, while bifidobacterial monoculture consumed almost all detectable substances, which is best illustrated by the electropherograms with FOS.

### 3.4. Coculture of Probiotics and Pathogen Strains on Different Fructans and Synbiotic Factor Assessment

In the study of the synbiotic activity of carbohydrates from burdock and Jerusalem artichoke and *B. bifidum* against *S. aureus*, coculture fermentation was carried out. The initial counts of bifidobacteria and staphylococci and the fermentation time varied. The fermentation time was varied to determine the optimal cocultivation time for the probiotics and pathogens in the range of 7 to 9 h based on the previously obtained growth curves of the monoculture and coculture. In the considered interval, definite concentrations of lactic and acetic acids must be achieved in order to change their effects on the pathogen from bacteriostatic to bacteriolytic.

With an equal initial count of bifidobacteria (7.95–8.02 log(CFU·mL^−1^)), after 7 h and 9 h fermentation, the final count at the lower time exceeded the value at higher time for similar substrates (Table 4). The closest values the final probiotic count following between 7 and 9 h of cultivation were achieved on Burd−20. At the minimal inoculum dose of *Bifidobacterium* approximately the same count of 7.43–7.53 log(CFU·mL^−1^) was reached after 8 h with all hydrocarbons except JA−80, where the lowest final bifidobacteria count was observed (7.09 log(CFU·mL^−1^)). It should be noted that in all variants of coculture fermentation on JA−80, the final bifidobacteria count was minimal relative to that of other substrates under the same conditions.

Despite the high final bifidobacteria count after 7 h of growth, the production of lactic acid with a lower *Staphylococcus* starting count was notably lower than the values obtained with a longer fermentation time and inoculum dose of *Staphylococcus*. However, these values were similar to those obtained during 8 h of growth of bifidobacteria monoculture (taking into account the close initial and final counts). It can be assumed that the production of lactic acid after 9 h of cocultivation was obtained as a result of coproduction of acids by bifidobacteria and staphylococci. However, it should be noted that the total acid content was higher than that in monocultures (which indicates different yields for mono and cocultures of bacteria). The production of acetic acid in 7-h co-cultures was the same as at 8 h, but in both cases, it was inferior to the data on the monoculture growth of bifidobacteria (the acetic acid concentrations on JA−80 and Burd−80 were closest to the monoculture values). This may be explained by the consumption of acetate by microorganisms. According to the obtained data, it can be seen that at a lower inoculum dose of the pathogen, the acid profile shifted towards acetic acid, which has a larger inhibitory effect on *Staphylococcus*. This indicates a positive influence on the synbiotic composition effectiveness.

The synbiotic factors were calculated using Equation (3). This criterion shows the magnitude of the decrease in the specific growth rate relative to the maximum value under the given conditions; the lower synbiotic factor means the lower specific growth rate. In the case, when the initial count of bifidobacteria (from 5.44 to 5.49 log(CFU·mL^−1^)) and the ratio with the initial count of *Staphylococcus* (about 1:28.3) were the smallest, the concentration of formed acids was not sufficient to inhibit the pathogen. The final count of *Staphylococcus* reached a value close to that of monoculture. In this case, the SF values differed slightly from each other in the range of 0.115 (FOS) to 0.156 (JA−20). On the contrary, with the minimum initial count of *Staphylococcus* (about 3.8 log(CFU·mL^−1^)) and the highest initial probiotic–pathogen ratio (about 1.5 × 10^4^ to 1), the final *Staphylococcus* count was significantly lower than the maximum, even with relatively low concentrations of the inhibitor. The growth of *Staphylococcus* was most strongly suppressed in FOS-containing medium, and the weakest suppression effect occurred in JA−80 medium. At the same time, the SF calculated for the FOS coculture was the smallest (–0.005), which indicates the greatest degree of suppression. The SF for the JA−80 coculture was the highest (0.176). Thus, the data obtained in the experiment are in agreement with the calculated SF values. With a moderate initial ratio of bifidobacteria and staphylococci (approximately 11.6:1), the greatest levels of pathogenic suppression were observed with JA−20 (SF 0.058) and Burd−20 (SF 0.089), and this was also correlated with the final pathogen count. The correlations between the synbiotic factors and integral values of the specific growth rates of staphylococci, calculated separately for each group of experiments, were strongly positive (see Table 4). However, the overall correlation was weak (r = 0.031), which can be connected to the difference in conditions. The use of the relative SF has been suggested previously. However, in the case of using the MIC model, SF can be equal to zero or negative. In this study, to compare the various conditions, the differences between SFs of tested and standard substances were calculated. These data are characterized by a strong positive relation (r = 0.858), which confirms the predictive value of the developed model.

## 4. Discussion

To solve the problem of food safety and its effects on human health, it is necessary to identify the patterns of growth suppression by food microbial contaminants and pathogens under the influence of physical or chemical factors (salt concentrations, temperature, pH, atmospheric oxygen, and others). Mathematical modeling of microbial growth is widely used to describe the growth of microorganisms that cause microbial food spoilage. Predictive microbiology was originally based on three main principles: (1) The growth, survival, and inactivation of microorganisms are considered reproducible reactions, (2) the behavior of microorganisms depends on a limited number of environmental factors; and (3) by means of quantifying the joint influence of these factors, the behavior of microorganisms can be predicted [33]. The most commonly used species used for modeling in this area are *Listeria monocytogenes*, *Salmonella* sp., *Staphylococcus* sp., *Bacillus cereus*, *Escherichia coli*, and some yeasts.

Models describing the influences of abiogenic factors are most widely used. The effects of sodium nitrite, sodium chloride, pH, and temperature and their interactions on the growth kinetics of *S. aureus* 196E have been described by quadratic and cubic polynomial models [34]. The Gompertz function was applied to generate growth curves. Quadratic models were used for the growth prediction of Salmonella spp., but pair interaction terms were excluded [35]. Based on these data, the Integrated Pathogen Modeling Program (IPMP) was developed to obtain growth curves depending on these factors [36]. This model allows the optimal parameters for the production and storage of food products to be selected, ensuring the growth of unwanted microorganisms. It is noted that these models only have good predictive ability if there is no limitation on any component of the media. This was also taken into account when forming the model in our study.

Models estimating the influences of biogenic factors (for example, organic acids formed by microorganisms during pickling or fermentation of foods) on the growth of pathogens are more suitable to fulfill the aim of our study. A square-root-type model was applied to simulate the growth of *Escherichia coli* at different pH levels and lactic acid concentrations (both dissociated and not dissociated) [37]. It was shown that maximum suppression of the *E. coli* growth was achieved by the combined effects of high concentrations of acids and a low pH; however, the concentration of nondissociated acid had a stronger effect. The square-root-type equation includes terms specifying the effect of temperature and water activity on the growth rate; thus, the biogenic and abiogenic factors can be combined [38]. In this case, the exponential phase growth rates from the data obtained by viable count assays were calculated by estimating the line of best fit, while the Gompertz function is more suitable for the turbidimetry data. Similarly, the specific growth rates in monoculture studies were calculated by estimating the line of best fit in our work.

A model containing the interaction of both abiogenic and biogenic factors as multiples of their functions was introduced [39,40]. Moreover, this equation included the minimal inhibitory concentration that has the biological value and shows the concentration of substances above which no growth occurs. Zuliani et al. [41] used this model to predict the influences of temperature, water activity, pH, and concentrations of lactic, acetic, and sorbic acid salts on the growth of *L. monocytogenes* in ground pork. The concentrations of nondissociated acids were taken into account, and the influence of pH on the inhibitory effect of the acid salts was established. A modified version of this model was applied in our study, with the functions taking into account temperature and water activity excluded from the equation, since these parameters were not varied. On the other hand, the inclusion of the MIC in the quantitative model allowed us to establish the relative effect of the probiotics on pathogenic growth as a result of organic acid formation. It should be noted that in the original equation, the MIC terms were zero when the acid concentration was greater than the MIC. However, this condition was not applied in our work, and it is possible that the specific growth rate could become negative. The introduction of a quasi-specific growth rate as the difference between the specific growth rate and the specific death rate [42] made it possible to associate the lysis of *Staphylococcus* observed in coculture with the increase in the death rate under the actions of the acids. This hypothesis, however, requires more detailed theoretical and experimental study.

The negative value of the specific growth rate can be interpreted as the bacteriolytic effect of bifidobacteria on *Staphylococcus aureus*. It was previously established [43] that at low pH values and under high concentrations of lactic acid, some lactic acid bacteria (LAB) strains can have a bactericidal effect on *S. aureus*. Moreover, the rate of staphylococci death under the influence of starter cultures in food matrices (cheeses) was lower, when the rate of pH decrease by starter cultures microorganisms was higher. In the current study, it was found that the increase in the initial count of *Staphylococcus* in coculture with bifidobacteria led to both a higher final count and to decrease in the death rate in the final stage. In other words, with an equal rate of acid formation by bifidobacteria but a greater initial count of *Staphylococcus*, the population can reach a larger final count before the growth rate takes a zero or negative value, and at a lower initial count, the opposite can occur. For more detailed study of the bacteriolytic action against *Staphylococcus*, the variant of longer fermentation was considered. In addition, there is a possibility that the combined inhibition effect of lactic and acetic acids may be higher than the combined effect of each of the acids separately.

In vitro activity of *B. longum* in synbiotics with FOS, GOS, frutalose, inulin, and arabinogalactan as pure substances or in mixtures with different ratios against enteropathogens (*E. coli*, *L. monocytogenes*, *Cronobacter sakazakii*, *Clostridium difficile*, *Salmonella enterica*, *Yersinia enterocolitica*, and *Shigella sonnei*) was assessed relative to the initial count of microorganisms [44]. It was found that despite GOS markedly stimulate the growth of bifidobacteria in monoculture the influence of probiotic to *E. coli* and *L. monocytogenes* in co-culture was not observed in medium with GOS, while the suppression of *C. difficile* was great (the final count in coculture was more than 4 log(CFU·mL^−1^) less than in the monoculture). On the other hand, our results showed that the same combination of probiotics and prebiotics can be more effective against the same pathogen at specific ratio of bacterial counts and less effective at the other ratios (the counts of *Staphylococcus* and SF for JA−20, Burd−20 and FOS at different initial bifidobacteria counts). In coculture of bifidobacteria and *C. difficile* in media with various prebiotics, it was found that the combination of *B. longum* and Actilight demonstrated the most potent bacteriolytic effect on the pathogens (the final count was lower than the initial one by approximately 0.6 log(CFU·mL^−1^) [45] However, the fermentation duration was 24 h, whereas in our study, it was limited to no more than 10 h. Possibly, a further decrease in the staphylococcal count could occur with a longer duration of coculture cultivation with JA−20, Burd−20, or FOS.

The dynamic nature of changes in parameters determining interspecific interactions in mixed cultures complicates their modeling. The formation of acids inhibiting pathogenic growth is associated with bacterial growth. Panebianco et al. [46] applied a predictive model based on the Lotka–Volterra equations to select and characterize LAB isolated from traditional dairy products produced in Calabria that are potentially usable as an adjunct culture against *L. monocytogenes* in cheese and to study their anti-Listeria activity in soft cheese during chilled storage. The model included the counts of LAB and *L. monocytogenes* at time t, the maximum growth rates, the maximum population densities, the interspecific competition parameters of LAB on *L. monocytogenes* and vice versa, and the physiological states of two populations. However, in this model, the measure of interaction is empirical coefficients that are not related to the physiology of microbes. Earlier [20], an approach characterizing the inhibitory effect of substances formed by one microorganism on another was proposed. This approach allowed us to combine the parameters (inhibition constants or MICs) in the model including the description of probiotic growth. It also allowed us to quantitatively measure the effectiveness of the synbiotic composition against the pathogen based on the model.

Regarding commercial prebiotics, a number of works have investigated the prebiotic activity of substances produced by extraction from plant raw materials. Jerusalem artichoke fructans were found to have a greater stimulating effect on *B. bifidum* than high molecular weight inulin [47]. The fructans obtained by water extraction from burdock roots followed by precipitation with 80% v/v ethanol had strong stimulating effects on the growth of *Bifidobacterium adolescentis* ATCC 15703 in vitro, and they also increased the counts of bifidobacteria and lactobacilli in mice [48]. Studies have been carried out to assess the prebiotic properties of burdock root flour as an ingredient in cookies [49], and these showed an increase *in Bifidobacterium bifidum* G90^®^ growth by 82% compared to a control treatment. In vivo it was established that the relative count of *Bifidobacterium* and *Lactobacillus* increased in an *A. lappa* group of mice compared with cellulose and commercial inulin groups. At the same time, a slight increase in the *Staphylococcus* concentration was detected in the inulin group compared with the *A. lappa* and cellulose groups [50]. However, as indicated in the study above with extracts, and even more so with plant flour, the effect on microorganisms may affect not only prebiotics but also growth inhibitors (for example, phenolic compounds). Additionally, different fractions of fructans (rich in FOS or high DP inulin) can have different prebiotic effects. It is known that precipitation with different concentrations of ethanol makes it possible to separate fructans into fractions according to the DP. For example, using an extract from *Echinacea* roots, it was shown that the molecules of fructans precipitated with 80, 60, and 40% ethanol were characterized by DPs of 33, 42, and 50, respectively [22]. In our work, it was shown that fructans of burdock and Jerusalem artichoke with higher DPs (precipitated by 20% ethanol) stimulate the antagonism of *B. bifidum* against *S. aureus* more strongly than commercial FOS and fructans of burdock and Jerusalem artichoke with lower DPs (precipitated by 80% ethanol).

The application of the proposed method for the estimation of prebiotic activity and SF seems quite promising, since it allows us to compare not only different plant raw materials and probiotics but also the isolation conditions as well as to reduce the number of groups in the final in vivo studies.

## 5. Conclusions

In this study, a model that describes the suppression of pathogenic bacteria by probiotics as a result of the inhibition of their growth with organic acids in coculture was proposed. The influence of the initial counts ratio of *Bifidobacterium bifidum* VKPM Ac−2136 and *Staphylococcus aureus* ATCC 43300 as well as the duration of fermentation to the competition outcome were considered. Investigation of the effectiveness of the synbiotic composition of *Bifidobacterium bifidum* and fructans isolated from *Arctium lappa* roots and *Helianthus tuberosus* tubers versus *Staphylococcus aureus* was carried out. The greatest suppression of pathogen growth by bifidobacteria was observed in combination with fructans from burdock precipitated with 20% ethanol as a carbohydrate substrate. In practical terms, the results obtained also allow to note that the combination of probiotic and prebiotic that demonstrate the greater inhibition of the pathogen growth at some initial bacterial count ratio, may be essentially less effective at the other ratio. Further research should be done to confirm the established patterns for fecal cultures.

## Figures and Tables

**Figure 1 microorganisms-09-00930-f001:**
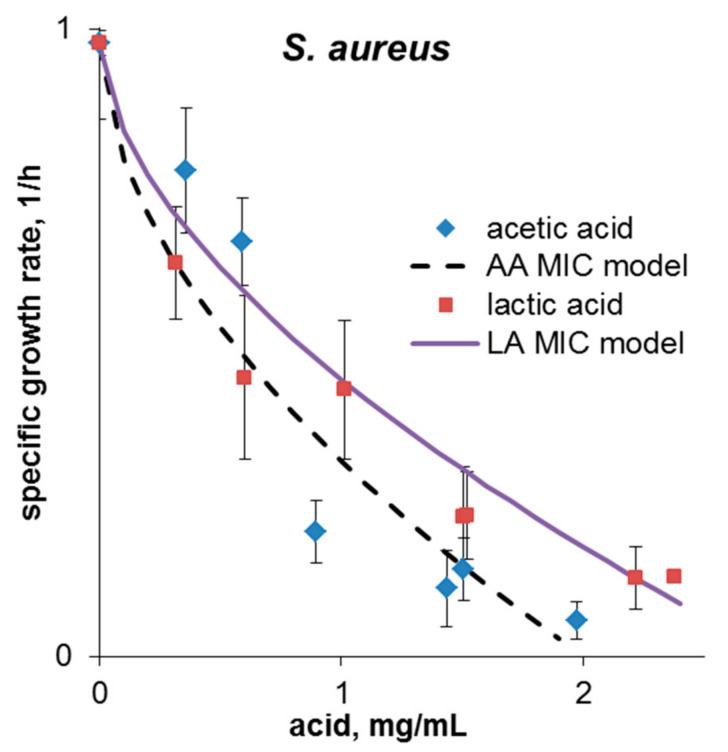
Curves *S. aureus* growth inhibition: Comparison of experimental and predicted data. Each specific growth rate value of the pathogen is the average of three measurements (*p* = 0.05).

**Figure 2 microorganisms-09-00930-f002:**
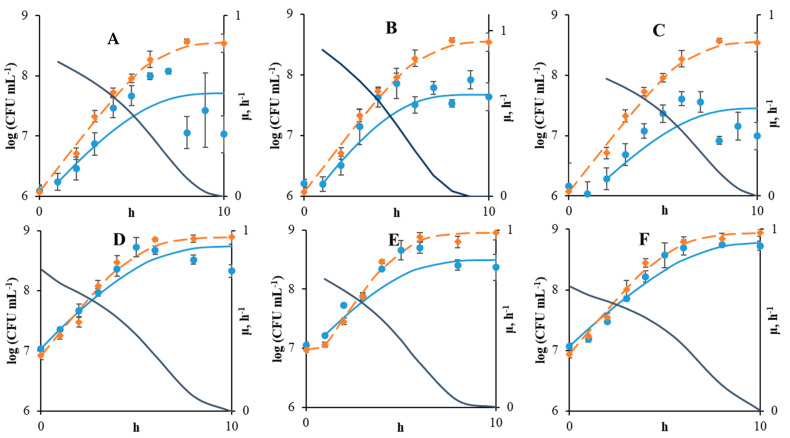
The experimental and calculated curves of *S. aureus* growth in monoculture (orange) and coculture (blue) at different initial cell counts ((**A**–**C**)-$x_{{Pt}_{0}}$ = 10^6^ CFU mL^−1^; (**D**–**F**)-$x_{{Pt}_{0}}$ = 10^7^ CFUmL^−1^) and with different substrates ((**A**,**D**)-glucose; (**B**,**E**)-FOS; (**C**,**F**)-lactulose). The curves of the calculated specific growth rates of the pathogen as a function of time (dark blue). To calculate the specific growth rate of the pathogen, the count of bifidobacteria was determined according to the exponential law. The concentration of lactic and acetic acids were defined using the yields values. To calculate the specific growth rate MIC equation was used. The growth curves were built using the calculation results from the MIC model in coculture and the Verhulst equation in monoculture.

**Figure 3 microorganisms-09-00930-f003:**
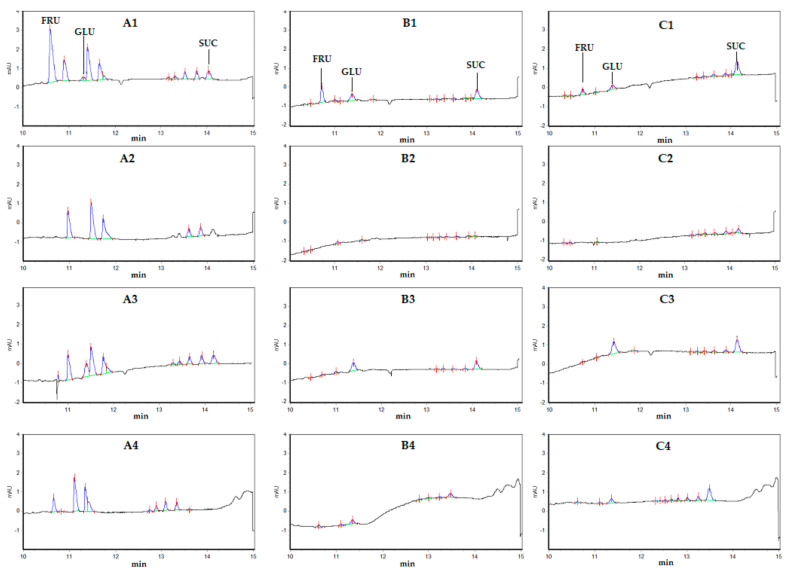
Electropherograms of FOS (**A**), Burd-20 (**B**) or JA-20 (**C**) samples; medium before inoculation (**1**) and cultural fluid after 8 h of fermentation of *S. aureus* monoculture (**2**), *B. bifidum* monoculture (**3**), and coculture (**4**). Standards were fructose (FRU), glucose (GLU), and sucrose (SUC). The left group of peaks is suggested to correspond to pure fructose oligomers and the right one to glucose-containing FOS.

**Table 1 microorganisms-09-00930-t001:** Inhibition constants of *S. aureus* monoculture and R^2^ values of regressions.

Acids	MIC (mg·mL^−1^)	α, β	R^2^
Lactic acid	2.8	0.66	0.94
Acetic acid	2.0	0.55	0.89

**Table 2 microorganisms-09-00930-t002:** Growth characteristics and parameters of the minimum inhibitory concentration (MIC) model calculated for mixed cultures of *S. aureus* and *B. bifidum* on glucose, fructooligosaccharides (FOS), or lactulose while varying the initial *Staphylococcus* population.

Substrate	Initial Count of *S. aureus*, log(CFU mL^−1^)	$\mu_{{Pr}_{max}}\;(h^{-1})$	$\mu_{{Pt}_{mrx}}\;(h^{-1})$	Acids Production, mg mL^−1^		Yield, pg CFU^−1^		SF	R^2^
				Lactic Asid	Acetic Acid	Lactic Acid YLX	Acetic Acid YAX		
Glucose	6.09 ± 0.10	0.47	0.89	2.13 ± 0.06	2.24 ± 0.07	0.37	0.39	−0.0015	0.93
	7.04 ± 0.04	0.48	0.76	1.96 ± 0.04	2.01 ± 0.04	0.40	0.39	−0.0016	0.99
FOS	6.21 ± 0.07	0.48	1.09	2.52 ± 0.05	2.34 ± 0.05	0.57	0.53	0.0003	0.80
	7.06 ± 0.05	0.50	0.81	2.25 ± 0.07	2.11 ± 0.06	0.60	0.56	0.0013	0.99
Lactulose	6.16 ± 0.38	0.48	0.81	0.77 ± 0.02	1.27 ± 0.03	0.17	0.28	−0.0026	0.86
	7.07 ± 0.05	0.50	0.66	0.78 ± 0.03	1.35 ± 0.03	0.15	0.27	0.0062	0.99

**Table 3 microorganisms-09-00930-t003:** The growth characteristics of monocultures *S. aureus* and *B. bifidum* on carbohydrate fractions of aqueous extracts of burdock (Burd) and Jerusalem artichoke (JA) precipitated with 20% or 80% ethanol and with control substrates (FOS and glucose (Glu)).

Microorganisms	Substrate	Bacterial Count, log(CFU·mL^−1^)		Acid Production, g L^−1^		Final pH
		0 h	8 h	Lactic Acid	Acetic Acid	
*Bif*. *bifidum*	JA−20	8.15 ± 0.06	9.03 ± 0.05	0.30 ± 0.02	0.96 ± 0.05	5.28
	JA−80	7.96 ± 0.01	9.21 ± 0.05	0.06 ± 0.00	0.25 ± 0.01	6.80
	Burd−20	8.16 ± 0.08	9.11 ± 0.05	0.05 ± 0.00	1.03 ± 0.03	5.12
	Burd−80	8.07 ± 0.04	9.47 ± 0.18	0.04 ± 0.00	0.33 ± 0.01	6.76
	FOS	8.17 ± 0.04	9.19 ± 0.05	0.56 ± 0.02	1.53 ± 0.03	4.57
	Glu	8.18 ± 0.04	8.75 ± 0.06	0.05 ± 0.00	1.06 ± 0.02	5.12
*S*. *aureus*	JA−20	5.89 ± 0.03	8.62 ± 0.04	0.74 ± 0.03	0.27 ± 0.01	6.18
	JA−80	6.10 ± 0.01	8.71 ± 0.07	0.34 ± 0.01	0.00 ± 0.00	6.78
	Burd−20	5.82 ± 0.08	8.67 ± 0.07	1.12 ± 0.03	0.41 ± 0.01	6.05
	Burd−80	6.19 ± 0.10	8.79 ± 0.05	0.41 ± 0.01	0.00 ± 0.00	6.74
	FOS	5.97 ± 0.03	8.58 ± 0.05	1.49 ± 0.03	0.32 ± 0.01	5.90
	Glu	6.14 ± 0.03	8.57 ± 0.11	2.11 ± 0.06	0.44 ± 0.02	5.62

**Table 4 microorganisms-09-00930-t004:** The growth characteristics of cultures and synbiotic factors for the carbohydrate fraction of aqueous extracts of Jerusalem artichoke and burdock precipitated with 20% or 80% ethanol and with the control experiments with FOS (Orafti P95) with varied initial counts of bifidobacteria and *Staphylococcus* and fermentation times.

Substrate	Fermentation Time, h	*B. bifidum* log(CFU·mL^−1^)		*S. aureus* log(CFU·mL^−1^)		Final pH	Acids Production, g L^−1^		*S. aureus* Integral Specific Growth Rate,	*SF*	r *	*SF* _dif_
		0 h	Final	0 h	Final		LA	AA	h^−1^			
JA−20	9	8.01 ± 0.08	9.12 ± 0.12	6.97 ± 0.12	7.62 ± 0.24	6.17	1.55 ± 0.03	0.46 ± 0.01	0.166	0.058		−0.053
JA−80		7.96 ± 0.10	9.08 ± 0.19	6.89 ± 0.13	8.60 ± 0.23	6.33	1.22 ± 0.02	0.37 ± 0.01	0.437	0.112		0.001
Burd−20		7.99 ± 0.13	9.34 ± 0.16	6.91 ± 0.13	7.19 ± 0.15	6.26	1.21 ± 0.02	0.56 ± 0.03	0.123	0.089	0.688	−0.022
Burd−80		7.96 ± 0.11	9.05 ± 0.14	6.88 ± 0.16	8.29 ± 0.19	6.42	1.06 ± 0.02	0.35 ± 0.01	0.360	0.146		0.035
FOS		7.96 ± 0.11	9.06 ± 0.11	6.91 ± 0.17	8.63 ± 0.14	6.31	0.96 ± 0.04	0.65 ± 0.03	0.440	0.111		0.000
JA−20	8	5.46 ± 0.10	7.51 ± 0.11	6.91 ± 0.15	8.72 ± 0.25	6.06	0.90 ± 0.03	0.14 ± 0.00	0.520	0.156		0.041
JA−80		5.47 ± 0.11	7.39 ± 0.11	6.90 ± 0.13	8.70 ± 0.20	5.96	0.90 ± 0.04	0.12 ± 0.00	0.518	0.143		0.029
Burd−20		5.46 ± 0.10	7.53 ± 0.12	6.93 ± 0.18	8.70 ± 0.16	6.22	1.07 ± 0.03	0.21 ± 0.01	0.509	0.141	0.685	0.027
Burd−80		5.49 ± 0.17	7.43 ± 0.11	6.91 ± 0.21	8.76 ± 0.13	6.00	0.89 ± 0.04	0.16 ± 0.01	0.532	0.144		0.030
FOS		5.44 ± 0.10	7.53 ± 0.11	6.92 ± 0.11	8.67 ± 0.17	6.07	1.04 ± 0.02	0.31 ± 0.01	0.503	0.115		0.000
JA−20	7	7.95 ± 0.15	9.45 ± 0.12	3.82 ± 0.33	6.66 ± 0.07	5.47	0.06 ± 0.00	0.31 ± 0.01	0.933	0.112		0.118
JA−80		7.96 ± 0.12	9.21 ± 0.15	3.70 ± 0.10	7.11 ± 0.03	5.67	0.06 ± 0.00	0.25 ± 0.01	1.120	0.176		0.181
Burd−20		8.02 ± 0.13	9.49 ± 0.16	3.86 ± 0.16	6.73 ± 0.05	5.29	0.16 ± 0.01	0.43 ± 0.02	0.943	0.052	0.854	0.057
Burd−80		8.07 ± 0.10	9.49 ± 0.18	3.88 ± 0.18	6.48 ± 0.08	5.43	0.04 ± 0.00	0.33 ± 0.01	0.854	0.103		0.108
FOS		7.95 ± 0.13	9.70 ± 0.15	3.70 ± 0.26	6.03 ± 0.10	5.02	0.23 ± 0.01	0.54 ± 0.02	0.766	−0.005		0.000

* r is the correlation between the *S. aureus* integral specific growth rate and SF for each group of experiments.

## Data Availability

Not applicable.

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
