# Peer review of "A Study on the Synbiotic Composition of Bifidobacterium bifidum and Fructans from Arctium lappa Roots and Helianthus tuberosus Tubers against Staphylococcus aureus"

_microorganisms, 2021, doi:10.3390/microorganisms9050930_

Round 1

Reviewer 1 Report

Dear Authors

Introduction summarizes relevant research to provide context and clearly state the problem.  The topics are well developed and confronted to other publications.

Methods are sufficient explained to replicate the research. In material and methods replace 75 ° C with 75 °C

The results are presented clearly and objectively, easily understood. In results Figure 2 are difficult. Maybe separate in more figures or with a different disposition in horizontal not vertical reading.

The discussion section interprets the findings in view of the results obtained in this and in past studies on this topic.

The reference list covers the relevant literature adequately and in an unbiased manner.

Author Response

Dear Reviewer,

thank you very much for your corrections. We tryed to correct everything. Pleas see comments in attached file.

Best regards,

Boris

Reviewer 2 Report

In this research article, the authors utilized in vitro and computational methods to characterize the antimicrobial potential of a symbiotic mix, comprised of Bifidobacterium bifidum and fructans.

There are some points that would require attention in the manuscript:

  1. Lines 16-17: the authors could opt for a more generalized statement at the first line of the abstract.
  2. Ιn Introduction, the authors should clearly state the aims of their study.
  3. In section 2.2 the authors should disclose the amount of raw material used for the extraction of fructans.
  4. The protocol for the co-incubations of the probiotic strain with aureus is not adequately described, the authors could describe it in greater detail to facilitate replication studies.
  5. Did the authors perform serial dilutions prior to agar plating (Section 2.4)?
  6. Figure 1. The authors should include indications of statistical significance if appropriate.
  7. Lines 638-642: could the observed effect be attributed to the fact that the pathogen cannot use fructans for an energy source? If so, that does not mean that fructans have antimicrobial effects against aureus. The authors should additionally clarify the protocol that led them to this conclusion.

Author Response

(The authors gave the same response as above.)
